# An Analysis of the Spatial Development of European Cities Based on Their Geometry and the CORINE Land Cover (CLC) Database

**DOI:** 10.3390/ijerph20032049

**Published:** 2023-01-22

**Authors:** Szymon Czyża, Karol Szuniewicz, Iwona Cieślak, Andrzej Biłozor, Tomasz Bajerowski

**Affiliations:** 1Department of Geoinformation and Cartography, Faculty of Geoengineering, University of Warmia and Mazury in Olsztyn, Prawocheńskiego 15, 10-720 Olsztyn, Poland; 2Department of Socio-Economic Geography, Faculty of Geoengineering, University of Warmia and Mazury in Olsztyn, Prawocheńskiego 15, 10-720 Olsztyn, Poland

**Keywords:** fractal dimension, urban development, urban and regional planning, CORINE Land Cover (CLC)

## Abstract

The study demonstrated that the rate of spatial development is correlated with its fractal dimension. The presented results indicate that the fractal dimension can be a useful tool for describing different phases of urban development. Therefore, the formulated research hypothesis states that the fractal dimension of cities’ external boundaries is correlated with the rate of spatial development in urban areas. The above implies that the higher the rate of spatial development, the smoother the external boundaries of urban investment. Rapidly developing cities contribute to considerable changes in land management, in particular in municipalities surrounding the urban core. Urban development processes often induce negative changes in land management and contribute to chaotic and unplanned development. To address these problems, new methods are being developed for modeling and predicting the rate of changes in transitional zones between urban and rural areas. These processes are particularly pronounced in urban space, whose expansion proceeds at an uneven pace. The aim of this study was to propose a method for describing urbanization processes that are based on the dependence between the urban growth rate, the fractal dimension, and basic geometric parameters, such as city area and the length of city boundaries. Based on the calculated changes in the values of these parameters, a classification system was proposed to identify distinctive phases of urban development. The study revealed that land cover databases are highly useful for such analyses. The study was conducted on 58 medium-size European cities with a population of up to 300,000, including France, Germany, Italy, Poland, and Croatia. The study demonstrated that the fractal dimension and the basic geometric parameters of urban boundaries are significantly correlated with the rate of the spatial development of cities. The proposed indicators can be used to describe the spatial development of urban areas and the rate of urban growth. The development of the analyzed cities was modeled with the use of CORINE Land Cover (CLC) data for 2000–2006–2012–2018, acquired under the EU Copernicus program.

## 1. Introduction

### 1.1. Spatial Development of Cities

Cities are the key areas in the development of every civilization, and they attract considerable attention in various fields of scientific inquiry [1,2,3]. However, urban development in Europe and the world is accompanied by negative phenomena. The accumulation of uncontrolled social, economic, and spatial processes exacerbates the scale of the encountered problems and affects a growing number of people residing in cities and the surrounding areas [4,5,6,7]. Increased mobility and unlimited access to mobile devices and the Internet contributes to the growth of cities and the urban population [8].

Urban development analyses indicate that urban expansion contributes to adverse changes in land use and management [9]. The rapid development of cities leads to urban crowding as well as land, water, and air pollution, which poses a public health risk [10]. Cities generally differ in structure and design, but upon closer inspection, certain similarities can be observed in urban forms [11,12].

Wilson, Ware, and Ware described three types of processes in the development of urban areas: infill development, expansion of the existing areas, and the creation of new areas that are located remotely from the urban core. The new settlements can be isolated, can be linear, or can form clusters [13].

One approach to identifying new urban areas postulates that infill development takes place when at least 40% of the transformed land is surrounded by urbanized areas [14]. Most of these areas are equipped with basic technical infrastructure, such as roads and public utilities, and with social infrastructure [13]. According to Ellman, the existing infrastructure is the main prerequisite for infill development [15]. This is a rational approach that accounts for the importance of infrastructure in the process of building cities. The emergence of new settlements puts pressure on the local authorities to deliver the required infrastructure [15,16]. From the point of view of landscape preservation and environmental protection, infill development contributes to the loss of open areas and urban ventilation corridors [17]. This approach confirms that urban development is strictly linked with a city’s geometry; therefore, it can be described with the use of geometric parameters such as surface area, a perimeter, and the fractal dimension.

Urban expansion takes place when the percentage of urbanized land surrounding a given area does not exceed 50% [18]. This approach focuses mainly on changes in land use, and urban expansion occurs when urban land-use types become predominant in each territory [19]. The conversion of undeveloped land to urban land can involve two processes [20]. The first process includes changes in the transitional zone, which is characterized by low availability of infrastructure. The second process involves changes outside the transitional zone, where new settlements are highly dispersed, and they are referred to as urban villages. Therefore, the existing land-use types are indicative of the stage of urban development, which implies that land cover data can also be used to analyze the spatial expansion of cities [21,22].

Urban expansion has far-reaching consequences for the functioning of entire ecosystems as well as the residents of urban, transitional, and rural areas [23]. This type of urban development is referred to as boundary or border development, where urban areas expand in parallel bands with an outer edge [1]. The above phenomenon is linked with the concept of the urban growth boundary (UGB), which has been extensively explored in the literature [24,25,26,27]. New settlements can be also remotely established from the existing urban centers [20]. The resulting changes in land use take place outside the transitional zone, and they are termed as leapfrog development [9]. The establishment of new settlements at a certain distance from the urban core is also known as isolated development, and it is characteristic of rural areas where new additions to the existing clusters of buildings match the local architectural style.

### 1.2. Fractal Dimension in Analyses of Urban Development

Urbanization is one of the most dynamic global processes. Urbanization drives social and economic growth, which is why cities continue to expand their area and population. Urbanization induces profound changes in space, and it gradually modifies land-use structures. Urban spaces and changes in areas that are directly subjected to urbanization pressure should be monitored. New analytical methods and techniques are needed to reliably assess the nature and rate of urbanization, in particular the spatial expansion of cities [28].

An analysis of the literature indicates that the boundaries of a city rarely coincide with its administrative boundaries and are difficult to define. The course and geometry of city boundaries play a key role in urban development, which is why they attract considerable research interest. Urban areas have a complex and extensive spatial structure, and the distance from the city center to city boundaries is difficult to determine. The boundaries of urban areas play a key role in urban research. However, the length of urban boundaries and the area enclosed by these boundaries cannot be objectively defined [29,30,31,32]. Therefore, the length of the urban boundary has to be accurately measured to determine the size of the urban population.

Increasingly accurate models of urban development are being developed to address progressing urbanization, which poses one of the greatest civilizational problems in the world. Research into local factors that promote urban development indicates that cities are self-organizing structures [33]. One of the most interesting approaches to analyzing urban development and changes in urban boundaries posits that cities should be regarded as fractal units [34]. A fractal is a geometrical figure whose individual parts are similar to the whole [35,36]. By definition, a fractal is a set for which the Hausdorff–Besicovitch dimension strictly exceeds the topological dimension [37,38]. A fractal is scaleless, and it cannot be described with traditional units such as length, surface area, volume, or density.

The fractal dimensions of cities are generally defined in two-dimensional space on the basis of digital maps and remote-sensing images [9,10,20]. If an urban fractal is defined in two-dimensional space, the urban area and the urban boundary can be described with the use of fractal dimensions. The urban boundary can be regarded as a fractal line [36,39,40,41]. A closed curve representing the urban boundary is defined as the urban periphery, where the Euclidean pattern of urban space can be described [37,42]. In practice, the upper boundary and the lower boundary of the fractal dimension of urban space are influenced by the method applied to define the study area. Two approaches can be used to derive the fractal time series of urban growth [43,44]. The first approach relies on a fixed study area [45] and the second approach on a variable study area [46,47].

In the literature, all fractal images represent prefractals, rather than true fractals in a mathematical sense. A true fractal has an infinite number of iterations that can be revealed only in the world of mathematics. A prefractal has a limited number of hierarchical levels with a fractal geometry. Cities are accidental prefractals rather than true fractals because the urban form cannot be described on a characteristic scale [48].

Two approaches have been proposed for determining the fractal dimensions of urban areas. In the first approach, fractal dimensions are calculated for the shape of a city’s boundary, whereas in the second approach, fractal dimensions are linked with the density of urban development [33]. These approaches imply that urban development resembles the growth of two-dimensional particle aggregates [49]. Several concepts have been proposed for modeling the spatial development of cities. A recent approach makes a reference to cluster expansion in terms of statistical physics, and it posits that new objects are not added to the cluster but rather are linked with it [50]. A model describing the development of urban settlements as miniature cities (subclusters) relies on the assumption that urban growth fuels further growth. According to the various approaches to urban development in the literature, urban growth should be regarded as a process of organic growth that begins in the urban core and spreads to miniature cities outside the urban core [28,51].

Spatial databases containing information about land cover as well as advanced techniques for processing and modeling spatial data are vast sources of knowledge, and they can be deployed to develop new tools for identifying and monitoring urban sprawl [52,53,54]. The CORINE Land Cover (CLC) database supports broad spatial analyses because the data describing land cover in Europe are characterized by spatial continuity and enable the nonambiguous identification of various land-use types. Most importantly, CLC databases are regularly developed, which facilitates analyses of the dynamics and rate of changes and supports forecasting. Other sources of data, such as the Urban Atlas and the Global Human Settlement Layer, are less versatile in this respect [15]. According to the literature, the CLC is a far more useful resource for small-scale studies, but it is a less reliable tool for analyses conducted on a larger scale [17]. The CLC project, initiated in 1985 and updated in 2000, 2006, 2012, and 2018, provides information on the EEA member countries (39 countries, EEA39). The minimum mapping unit (MMU) in the CLC is 25 hectares (ha) for aerial phenomena and a minimum width of 100 m for linear phenomena. Land cover is mapped mainly on the basis of the visual interpretation of high-resolution satellite images. The CLC data set consists of 44 land cover classes grouped into five main categories: artificial surfaces, agricultural areas, forest and seminatural areas, wetlands, and water bodies [55,56,57]. Studies that rely on CLC data have certain limitations, such as the detailed nature of input data and interpretation methods and, consequently, a high degree of generalization. In areas characterized by considerable land fragmentation, the results can be generalized into dominant land-use types, which leads to a certain loss of information.

In this study, the development of urbanized areas was described and modeled with the use of the fractal dimension calculated on the basis of CLC data. The main aim of this study was to propose a method for describing the spatial development of a city. Basic geometric parameters were computed in selected European cities, and changes in these parameters were analyzed over time to propose a formula for describing and classifying cities on the basis of their stage of evolution. This research is important because of the importance of monitoring urbanization processes. The adopted method makes it possible to identify the trends and dynamics of the processes taking place, providing an important indicator for optimizing the spatial policy of the cities under analysis. The proposed procedure may contribute to extending the scope of research in spatial terms (selection of analyzed cities), in temporal terms (successive time intervals), and in terms of indicators (the analysis may be extended by adding subsequent indicators, e.g., number of inhabitants). The application of new elements in the analysis may contribute to the enrichment of empirical research in the identification of urban development processes. The spatial expansion of cities, the fractal dimension, and CLC data are described in the Introduction. The cities selected for the study and the procedure for calculating the fractal dimensions of city boundaries and the rate of urban development are described in the Section 2. The indicators calculated for the analyzed cities, the increase in the urban area, and the fractal dimension of the analyzed cities are presented, and the examined cities are classified on the basis of the stages of urban expansion in different time intervals in the Section 3. The results are discussed in the Section 4, and the recommendations for future research are formulated in the Section 5.

## 2. Materials and Methods

### 2.1. Study Area

The rate of spatial development was analyzed in 58 European cities, which were selected for the study on the basis of the availability of CLC data for 2000, 2006, 2012, and 2018 (Figure 1). The evaluated medium-size cities had a population of up to 300,000 at the beginning of the analyzed period. Owing to considerable differences in the social and economic development of European cities, only cities in the European Union, including France, Germany, Italy, Poland, United Kingdom, and Croatia, were selected for the study. In line with the New Urban Agenda [58], the study was conducted in European cities with the greatest potential to make Europe a global reference point for identifying, experimenting, and applying solutions to future urban challenges [59]. Land cover data for the examined cities were acquired from the CLC database for 2000, 2006, 2012, and 2018. CORINE Land Cover data are publicly available, and they are stored and managed by the respective national authorities in the EU [60]. In addition, the use of data from the CLC database, owing to the different levels of accuracy in the data sets, where older data sets have less accuracy than newer ones, required visual verification of the different land-use types occurring for each urban area. The authors in this paper wanted to accurately determine geometric indicators for urban areas, so they focused on 58 European cities, and they bore in mind that historical CLC data sets have some limitations, such as the detailed nature of the input data and methods of interpretation.

The cities located in different European countries and characterized by various spatial attributes were compared to verify the fractal dimension as an indicator for evaluating the rate of urban development. The analyzed set included coastal cities (Aberdeen, Rijeka, Trieste) as well as inland cities (Cheltenham, Strasbourg, Crawley). The limiting effect of rivers’ intersecting urban areas was also considered. In the past, rivers played an important role in the establishment of urban settlements, and they presently pose a significant barrier to the expansion of local transportation systems. Therefore, cities intersected by rivers (Verona, Norwich, Nantes) as well as cities that were not built around water courses (Oviedo, Białystok, Erfurt) were included in the analysis. The social, economic, legal, and spatial aspects of urban development differ across European countries; therefore, the evaluated cities most probably differ according to other distinguishing features. The fractal dimension was determined to objectivize the process of describing urban development.

The available sources of land cover data were analyzed for the needs of this study. The CLC inventory was selected thanks to its availability, broad coverage, and confirmed usefulness in evaluations of urban development [61,62,63,64]. The boundaries of urban areas had to be determined in detail; therefore, land use was described on the basis of level 3 CLC data. The land cover map for selected cities is presented in Figure 2. In the studied group of 48 cities, land cover was determined at two points in time on the basis of CLC data for 2000, 2006, 2012, and 2018. This approach supported the identification of significant changes in the fractal dimension of city boundaries. The boundaries of the studied cities were identified in detail, which facilitated analyses of the rate of urbanization.

### 2.2. Procedure for Calculating the Fractal Dimension of Boundaries and the Rate of Urban Development

This study was inspired by the referenced works and by research into the dependence between the fractal dimension of the external boundaries of various land-use types and the degree of anthropogenic pressure [64].

For the needs of the study, urban areas were selected by determining which of the 58 land cover classes in the CLC inventory should be used for further analysis. The main land cover classes characteristic of urban areas were described on the basis of the principles for identifying urban morphological zones (UMZs). In the CLC inventory, level 1 landscape patterns represented artificial surfaces. In some cities, level 3 patterns denoting forests and seminatural areas were also classified as urban areas. This was the case when forests and seminatural ecosystems were surrounded in their entirety by artificial surfaces. The selected areas were then aggregated into a single group. In the following step, urban boundaries were verified on the basis of high-resolution satellite images for the analyzed time periods [65]. The procedure was repeated for CLC 2000, 2006, 2012, and 2018 data. The urban boundaries identified on the basis of CLC data for 2000 2006, 2012, and 2018 were compared to identify changes in urbanization processes in the evaluated cities (Figure 3).

In this study, the fractal dimension of urban development was analyzed on the basis of research into land cover and land-use types [51,66]. The fractal dimension of urban areas was determined with the box-counting method. The hierarchical grid method proposed by De Cola was used, where a grid of boxes is superimposed on a fractal, and the objects intersected by the fractal are counted [67]. In this approach, the number of boxes represents the fractal’s surface area, and the combined length of box edges is the fractal’s perimeter. The size of the boxes in the grid overlay is decreased in successive iterations. The box-counting method is used to identify changes in the fractal’s perimeter when the length of the box edges is decreased [68]. In urban structures, the number of elements in successive iterations is not constant; therefore, the box-counting dimension is defined as a boundary value where the length of the box sides decreases toward zero. According to Equation (1),
(1)$$O=c*{\sqrt{S}}^{D_{f}}$$
where *O* is the perimeter;*c* is the shape constant;*S* is the surface area;*D_f_* is the box-counting fractal dimension.

Therefore, the box-counting dimension is defined by Equation (2):(2)$$D_{f}=\frac{\log O}{\log\sqrt{S}}$$

The box-counting dimension is determined as the slope of a regression line in a graph presenting the correlations between log values. In practice, real-world objects are usually modeled with the box-counting method and the compass-walking method. The relevant technique has to be selected when planning the experiment because it will influence the preparation of input data and the choice of calculation methods.

The obtained results were compared with CLC data, which were acquired under the Copernicus program. The CLC inventory was used because not all urbanized areas are contained within the administrative boundaries of cities. As a result, these areas are not taken into account in analyses of the spatial development of cities. There is no single harmonized definition of a city and a functional urban area, which considerably impedes analyses of European cities [19]. For this reason, the boundaries of urban areas were identified by introducing the urban morphological zone (UMZ) concept, which denotes continuous urban fabric with different population density. An urban morphological zone can be defined as “a set of urban areas laying less than 200 m apart”. These urban areas are defined on the basis of land cover classes describing the urban tissue and function [69]. The UMZ database was created by the European Environment Agency (EEA) on the basis of CLC data and automated methods for defining the boundaries of urban agglomerations. Maximum distance is the key criterion. Urban morphological zones comprise areas of dense urban development (separated by a maximum distance of 200 m), and they consist of the following land cover classes: continuous or discontinuous urban fabric, industrial or commercial units, green urban areas, selected forest areas, port areas, airports, sport and leisure facilities, and road and railway networks [70]. The adopted solution for identifying urban areas relied on a set of spatial data developed on the basis of the CLC inventory and the assumptions for defining UMZs [71].

The calculation of the fractal dimension and the determination of the rate of urban development in the analyzed cities between 2000–2006, 2006–2012, and 2012–2018 were important steps in the adopted procedure. A standard urban development model was adopted to describe population growth and human behaviors relating to optimal decision-making within the allocation of time and according to required effort and resources. There are many examples linking the applied model with economic, social, and health-related behaviors, including descriptions of the exponential growth of urban populations [72,73].

The fractal dimension (*D_f_*) was calculated with the box-counting method, also known as the area-perimeter method. In the adopted approach, the variables were the surface area of the fragments of the grids covering the studied object (*S*), which is described by the number of boxes, and the object’s perimeter (*O*), which is expressed by an equal number of box edges in a given fragment of the grid. All calculations were performed in the QGIS program. A geographic database containing information about land-use types in the analyzed years and classes of objects representing grid boxes with specific dimensions was generated. An algorithm was created for calculating the surface area and the perimeter of grid boxes. The fractal dimension was calculated on the basis of the values computed for boxes with a minimum surface area of 625 m^2^ and a maximum surface area of 625 km^2^. The box area was determined in 30 steps. The algorithm was tested and applied to boxes in each size category on the basis of CLC data for 2000, 2006, 2012, and 2018. The increase in the fractal dimension *(*Δ*D_f_*) was calculated with the use of Equation (3):(3)$$\Delta D_{f}=\left(\frac{D2_{i}-D1_{i}}{D2_{i}}\right)$$
where Δ*D_f_* is the increase in the fractal dimension;*D*1 is the fractal dimension in time *t*;*D*2 is the fractal dimension in time *t +* 1;*i* is the number for the analyzed city.

The increase in the fractal dimension of cities’ external boundaries is correlated with the rate of spatial development in urban areas. Higher values of increase in the fractal dimension imply a higher rate of spatial development, creating a ragged and dendritic urban fabric. On the other hand, the lower values of the increase indicator are related mostly to filling the areas between the main traffic routes and areas vulnerable to the pressures of urbanization processes while avoiding natural and anthropogenic obstacles. The study was expanded to include an analysis of changes in the area and perimeter of geometric figures that represent urban land-use types and were used to calculate the fractal dimension. The rate of changes in figure geometry and the fractal dimension was determined in successive time intervals: 2000–2006, 2006–2012, and 2012–2018.

The spatial development of cities (increase in urban area) in each time interval (2000–2006, 2006–2012, and 2012–2018) was calculated with the use of Equation (4):(4)$$\Delta A_{i}=\left(\frac{A2_{i}-A1_{i}}{A2_{i}}\right)$$
where Δ*A*
_i_ is the urban growth rate *i* city;*A*1 is the urban area in time *t*;*A*2 is the urban area in time *t* + 1,*i* is the number for the analyzed city.

The values of the urban growth $\Delta A_{i}$ rate indicator can range from 0 to several points. Values close to 0 indicate that a city’s area did not increase in the analyzed period, which implies that its development was inhibited. This is often the case when urban development is constrained by natural or artificial barriers. Values higher than 1 are indicative of dynamic urban development.

The increase in the perimeter of the geometric figure, i.e., the increase in the length of city boundaries, was calculated with the use of Equation (5):(5)$$\Delta P_{i}=\left(\frac{P2_{i}-P1_{i}}{P2_{i}}\right)$$
whereΔ*APi* is the increase in the length of *i* city boundaries;*P*1 is the boundary length in time *t*;*P*2 is the boundary length in time *t* + 1;*i* is the number for the analyzed city.

The increase in boundary length denotes the degree of figure filling. Ragged boundaries are longer, and they could imply that a rapidly developing city has annexed the surrounding areas. In turn, smoother boundaries are shorter, which suggests that the urban fabric is more compact and urban growth relies mainly on infill development (for example, in vacant areas between transport routes).

In order to organize the steps taken in the procedure for determining the phases of urban development, we can identify three stages. The first involves the determination of the study area. Within the framework of the activities undertaken, it is necessary to make a selection of cities and characterize the data sets that allow the identification of urban processes. The second stage involves data analysis and the determination of boundaries and classes characteristic of urbanized areas. The next step assumes the calculation of basic geometrical parameters (area, perimeter) and the fractal dimension. According to the acquired data, an increase in the above indicators is determined for the available time intervals. The last step involves the verification of the obtained results and the determination of a pattern for classifying the stages of urban development for every city. The respective increases in the fractal dimension *(*Δ*D_f_*), area (Δ*A*), and boundary (perimeter) length (Δ*P*) were calculated in the next step. The relationships between the calculated values were analyzed in all time intervals, but significant correlations were not observed. Therefore, it was assumed that the calculated increases can provide additional information on spatial expansion. An urban expansion matrix was developed on the basis of the calculated increase in the analyzed parameters and literature data (Table 1). The examined cities were assigned to different urban expansion classes on the basis of the calculated changes in perimeter, area, and the fractal dimension. All possible change combinations in 2000–2006, 2006–2012, and 2012–2018 were determined, and the increase (↑) and decrease (↓) in the analyzed indicators were calculated. The resulting urban expansion matrix was composed of eight classes, and it was used to determine the rate of spatial changes in each city. The classification process based on the urban expansion matrix is presented in Table 1.

On the basis of the calculated changes in perimeter, area, and the fractal dimension, the analyzed cities were divided into eight classes, denoting different phases (stages) of spatial expansion. The proposed classes are described below:Class 8—in this rapid urban expansion, the city expands in an uncontrolled manner in all directions. New urban fabric is ragged and dendritic.Class 7—urban expansion involves mainly infill development between major transport routes, and obstacles are bypassed. Areas that are relatively sensitive to urbanization pressure are annexed by the city. City boundaries are more compact and regular.Class 6—urban expansion takes place only in the vicinity of urban infrastructure. Urban boundaries are smoother, but the city has a dendritic shape.Class 5—urban expansion takes place in areas that are more resistant to anthropogenic pressure. Urban boundaries are smoother and more compact.Class 4—urbanized areas disappear. City boundaries are more ragged and dendritic in shape. Examples of the above include reclaimed areas with urban infrastructure, in particular linear infrastructure.Class 3—in urban regression, urbanized areas disappear from the urban periphery. Urban boundaries are compact but ragged.Class 2—in the regression phase, urbanization regresses, and urbanized areas are found mainly in the vicinity of linear infrastructure. Urban boundaries evolve into a dendritic shape.Class 1—full regression occurs in all directions.

The developed classification system was used to describe the phases of urban development in the examined cities in 2000–2006, 2006–2012, and 2012–2018. The proposed method was applied to examine the intensity and rate of changes in urban development.

## 3. Results

The study was conducted in 58 European cities. Their surface area, perimeter, and fractal dimension (*D_f_*) calculated on the basis of CLC data for 2000, 2006, 2012, and 2018 are presented in Table A1. The urban growth rate indicator was calculated for all cities on the basis of the increase in their area, perimeter, and fractal dimension in 2000–2006, 2006–2012, and 2012–2018 (Table 2).

In terms of the increase in city area (Δ*A*), the values of the urban growth rate indicator considerably differed across the examined time intervals. In 2000–2006, the greatest increase in urban area was noted in Verona (0.673), Alicante (0.609), and Rijeka (0.480). In 2006–2012, the greatest increase in urban area was observed in Monchengladbach (0.578), Chemnitz (0.346), and Almeria (0.2410). These cities are popular tourist destinations with similar locations and geographical features, which suggests that these factors played a role in their development in the analyzed periods. Similar observations were made in the last time interval. In 2012–2018, the greatest increase in urban area was noted in Västerås (0.353), Burgos (0.323), and Tampere (0.050). An analysis of changes in the fractal dimension (Δ*D_f_*) over time produced equally interesting results in 2000–2006: the greatest increase in the fractal dimension was observed in Burgos (0.102), Vigo (0.056), and Trieste (0.041). In 2006–2012, the greatest increase in the fractal dimension was noted in Munster (0.020), Santander (0.012), and Osijek (0.010). In 2012–2018, the greatest increase in the fractal dimension was observed in Västerås (0.061), Burgos (0.030), and Messina (0.009). The greatest increase in city perimeter (*ΔP*) was noted in Vigo (1.338), Verona (1.216), and Burgos (0.946) in 2000–2006; in Chemnitz (0.369), Monchengladbach (0.362), and Santander (0.242) in 2006–2012; and in Västerås (2.557), Burgos (0.274), and Verona (0.106) in 2012–2018. Moreover, the greatest increase in the analyzed parameters was observed in cities that are popular tourist destinations and, in two cases (Chemnitz and Munster), in rapidly growing industrial centers. In some cases, the analyzed parameters increased at a steady rate, which confirms the hypothesis that urban expansion proceeds at an uneven rate.

In line with the adopted procedure, the analyzed cities were divided into eight classes describing different phases (stages) of spatial development in the examined time intervals. The examined cities were allocated to different classes on the basis of the previously calculated increase (↑) or decrease (↓) in city area, perimeter, and the fractal dimension in 2000–2006, 2006–2012, and 2012–2018 (Table 1 and Table 2). The resulting classification of the analyzed cities is presented in Table 3.

The proposed classification system presents the stages of urban expansion in 58 European cities in 2000–2006, 2006–2012, and 2012–2018. Rapid changes in the rate of spatial development can be observed across the examined time intervals. Some cities were assigned to radically different classes in each period, including Alicante (8-1-8), whereas a rapid decrease in the rate of spatial development was observed in Split, Częstochowa, and Toruń (7-8-1). Between 2000 and 2018, a high and steady rate of urban development was observed in Białystok, Derby, Freiburg, Rijeka, Northampton, and Norwich, where the values of all examined parameters increased. Aberdeen, Almeria, Erfurt, Geneva, Trieste, Trondheim, and Verona were characterized by a relatively steady rate of urban expansion, and none of these cities moved up or down by more than one or two classes in the analyzed period. In the remaining cities, spatial expansion proceeded in rapid spurts. The phases of spatial expansion in the studied cities are presented in maps in Figure 4.

**Figure 4 ijerph-20-02049-f004:**
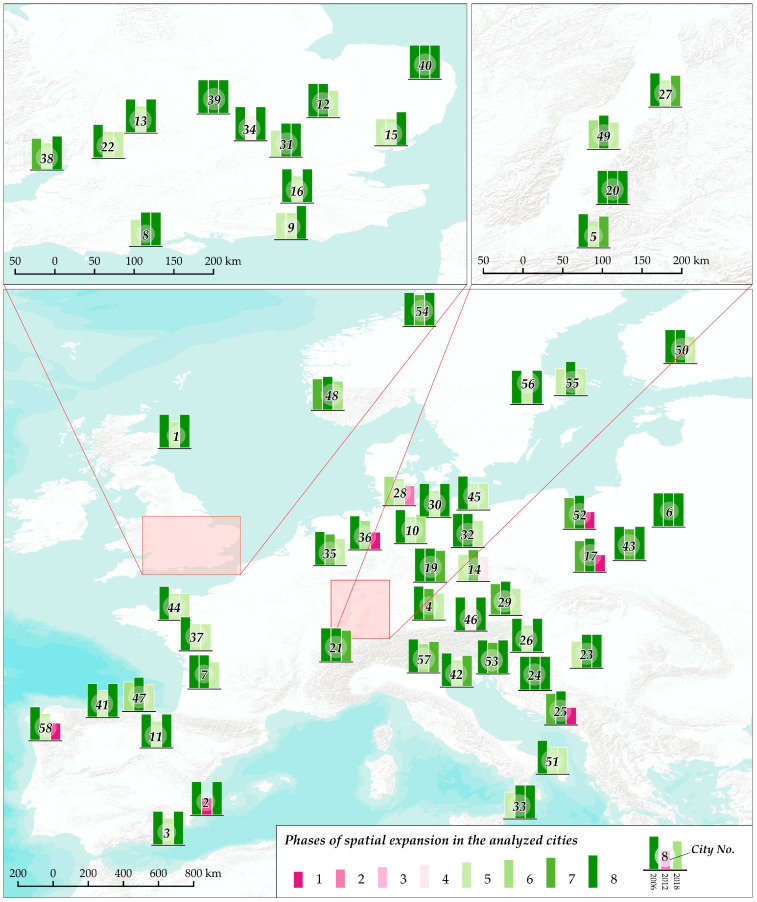
Phases of spatial expansion in the analyzed cities, labeled as in Table 3 and Table 1.

## 4. Discussion

Geospatial data, including CLC data, are a rich source of knowledge, and they can be used to develop new tools for identifying and monitoring urbanization processes. In areas that are most exposed to urbanization pressure, the degree and rate of urban expansion can be most effectively monitored by analyzing land cover data. However, this approach also has many limitations, which became apparent at the stage of calculating the boundaries and the fractal dimension of the cities.

The maximum city area for calculating the fractal dimension in the box-counting method was set at a maximum side length of 25,000 m, and the minimum side length was set at 25 m. In the calculations, 30 intermediate values of box side length (min-max) were adopted. Box sizes were compared in all evaluated periods. The comparison for 2018 was based on Sentinel data, which are more accurate (to the nearest 10 m) than other data sets. However, the acquired data were analyzed only in the vector format on the assumption that CLC data for 2000, 2006, 2012, and 2018 can be used to analyze changes in the surface area of the studied cities. Urban morphological zones were established for the needs of the analysis. However, a detailed analysis of the data set prepared on the basis of UMZ guidelines revealed that despite the absence of changes in land use, the same areas were differently classified in the CLC database in the studied periods. For example, the classification of undeveloped areas changed in Alicante (black circle in Figure 5). Undeveloped areas in the CLC database were marked with code 3.2.3 (sclerophyllous vegetation; 3.2 scrub and/or herbaceous vegetation associations; and 3 forest and seminatural areas) (Figure 5a) in 2000 and with code 3.2.1 (natural grassland) in 2006 (Figure 5b).

According to the research conducted, it can be concluded that there are specific cases such as Alicante that require in-depth analysis. The background of the changes that have taken place is related to the development of transport infrastructure and the temporary identification of adjacent areas as built-up areas. The consequence of the processes that have occurred is the infilling of land between the main traffic routes. Such processes affect the dynamics of change in terms of urban development phases. In view of the above, the authors believe that it makes sense to continue the research and extend it to include further time scales that would allow a more accurate identification of the urbanization processes.

In Lubeck, the area marked with code 1.1.2 (discontinuous urban fabric; 1.1 urban fabric; 1 artificial surfaces) in 2000 (black circle in Figure 6a) was marked with code 3.1.1 (broad-leaved forest; 3.1 forest; and 3 forest and seminatural areas) in 2006 (black circle in Figure 6b).

In Milton Keynes, the area marked with code 2.3.1 (pastures) in 2000 (black circle in Figure 7a) was classified as an urban area in 2006 (code 1.4.1 green urban areas) (black circle in Figure 7b) despite the absence of any changes in land use in this period.

These observations suggest that the fractal dimension, area, and perimeter of objects classified as urban areas could have been considerably influenced by the image interpretation method adopted in the selected periods.

The update frequency of land cover data in the vector format is yet another limitation in analyses on urbanization processes. The update cycle for CLC is 6 years. This update frequency appears reasonable for urbanization studies, but it significantly affects the time scale of the analysis. On the basis of the quality and specificity of the available information, land cover data can be reliably compared beginning from 2000, which implies that only four data sets can be used for analysis. This is a relatively small sample, in particular in studies that explore dynamic urbanization processes. The quality of the results is also affected by rapid technological advancements in data acquisition methods, which affects the accuracy of the available information. The above could pose a considerable problem, in particular in comparative analyses.

The rate of urban growth is determined mainly by the urban morphology (urban form) and its relationship with the size of a city, its functions, and the economic, social, technical, and environmental determinants of development. Observations of long-term, continuous urbanization processes generate valuable and sometimes-surprising results. In many cities, the urban growth rate indicator assumed negative values in the studied period. This could imply that many areas that are influenced and transformed by urbanization processes ultimately evolve into nonurban forms. The above suggests that urban development is a more dynamic and bidirectional process than was previously thought. Urban expansion is not always represented by positive values of the urban growth rate indicator, and negative values can be observed in certain stages of the urban life cycle [74]. Urban development or expansion begins in land that is most accessible. Therefore, urban land-use types can be compared to a predator in Lotka–Volterra equations, and they aggressively dominate over the remaining (surrounding) land-use forms [54,55,56]. In this analogy, “weaker” land-use types are more rapidly transformed into the dominant land-use forms. The results of the analysis presented in the above tables produce interesting conclusions concerning the rate of urbanization in the evaluated cities. Cities with the highest rate of urban growth in the analyzed period were in Germany (Freiburg im Breisgau), Croatia (Grad Rijeka), the United Kingdom (Derby, Northampton, and Norwich), and Poland (Białystok), a finding that is largely consistent with the economic trends reported in these countries in the studied period.

Most of the analyzed 58 cities were characterized by positive values of the urban growth rate indicator in each of the three studied time intervals. Negative indicator values were observed in only five cases (in total, 174 indicator values were calculated) in the examined period. This result confirms that urbanization exerts considerable anthropogenic pressure, and areas affected by urbanization rarely regress to other land-use types. The above could also imply that area expansion is not a highly reliable indicator for analyzing the development of areas that have already undergone urbanization. The territorial expansion of cities could be more effectively examined on the basis of the increase in the length of their boundaries and the fractal dimension, which, as demonstrated by this study, are characterized by much greater variation.

The proposed classification method effectively identifies trends in the spatial development of cities [75]. The number of classes is directly associated with the number of parameters adopted for the analysis. The phases (stages) of urbanization and changes in the spatial development of cities were described by assigning cities to different classes on the basis of changes in their area, perimeter, and fractal dimension. High and sudden increases in perimeter and the fractal dimension denote the rapid and often-uncoordinated spatial development of urbanized areas.

## 5. Conclusions

Analyses of the rate of urban development play key roles in land management in the context of monitoring the growth of cities and the adjacent areas and undertaking preventive measures. The rate of urban development can be accurately evaluated with the use of spatial data. The proposed procedure for evaluating the spatial development of cities requires access to regularly updated land cover data. Land cover data sets and advanced techniques for processing and modeling spatial data are rich sources of knowledge, and they can be used to develop new tools for identifying and monitoring the spatial development of cities. The CLC inventory is characterized by spatial continuity, and it can be reliably used to identify various land-use types and analyze the geometry of cities. CORINE Land Cover data are acquired at regular time intervals; therefore, changes in land use can be reliably monitored. Several key conclusions can be formulated on the basis of the analysis of the external boundaries of urban areas.

The proposed classification of the development phases of medium-size European cities indicates development trends and can be used as a tool for monitoring urban urbanization. On the basis of the analyzed cases, the stability of development of cities in the central part of Europe can be observed. Additionally, we can find individual cases, such as Munster, Vigo, Kiel, and Augsburg, where there is a steady decline in development phases. The authors point out that the usefulness of the tool will be even greater in the future, when it will be possible to carry out analyses for further time intervals. The method adopted also makes it possible to identify specific cases of changes in urban development phases, using Alicante as an example (grades 8-1-8). Such cases require in-depth analyses. It should be noted that they may be due to the processes of intensive development of infrastructure investments taking place. Consequently, in their surroundings, many areas act as servicing areas for the investments, thus being classified as urbanized areas. On the other hand, after the investment has completed, they return to their original uses, being identified as nonurbanized areas. However, in the long term, the completed investment over time induces urbanization processes in the neighboring areas. Consequently, neighboring areas are again identified as urbanized land-use classes.

The results of the conducted analysis indicate that the CLC inventory is a useful resource for describing the rate of urban growth. A high rate of urbanization leads to rapid development regardless of the attributes of built-up land, which can speed up or slow down this process to a varied degree. These findings could suggest that slow urbanization promotes the development of areas with optimal spatial attributes.

The results of this study and the derived conclusions can be used to formulate a preliminary rule concerning the evolution of the external boundaries of cities: cities where urban development proceeds at a faster rate are characterized by smoother external boundaries. Therefore, cities (urban morphological zones) whose spatial development proceeds at a faster rate tend to have ragged external boundaries. The reverse also applies: the fractal dimension of cities’ external boundaries is higher in urban areas characterized by a lower rate of spatial development.

The study demonstrated that urbanization not only increases the area of urbanized areas but also leads to changes in the length and shape of city boundaries. These geometric parameters could play key roles in describing the intensity of urbanization processes. The proposed procedure can be applied in the preliminary stage of identifying the phase of urban development. The results can be used to determine whether cities evolve in a planned manner and whether spatial transformation processes lead to the creation of optimal, compact urban forms.

The identification of the directions and rate of the urban development in Europe and in other parts of the world poses a considerable challenge. Information on potential land use is key to enhancing inclusive and sustainable urbanization. Urban sprawl and inefficient use of land continue to pose a problem, with varying impacts in different contexts. Modern technologies, such as satellite data, support the continuous monitoring of the changes in, standardization of, and protection of citizens’ privacy. Urban planning solutions that make optimal use of the available space will maximize social benefits, support the identification of areas that require careful regulation, and promote forward-looking urban growth strategies.

## Figures and Tables

**Figure 1 ijerph-20-02049-f001:**
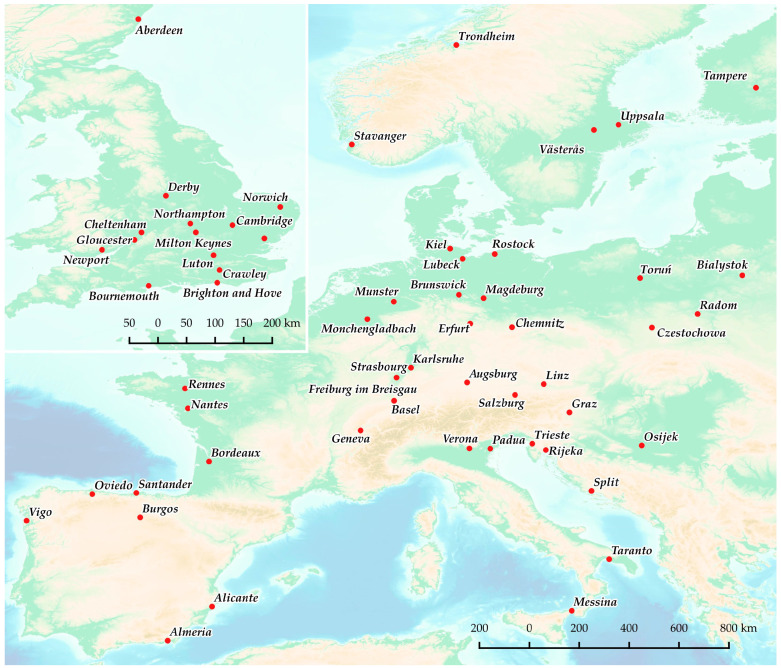
Location of the analyzed cities.

**Figure 2 ijerph-20-02049-f002:**
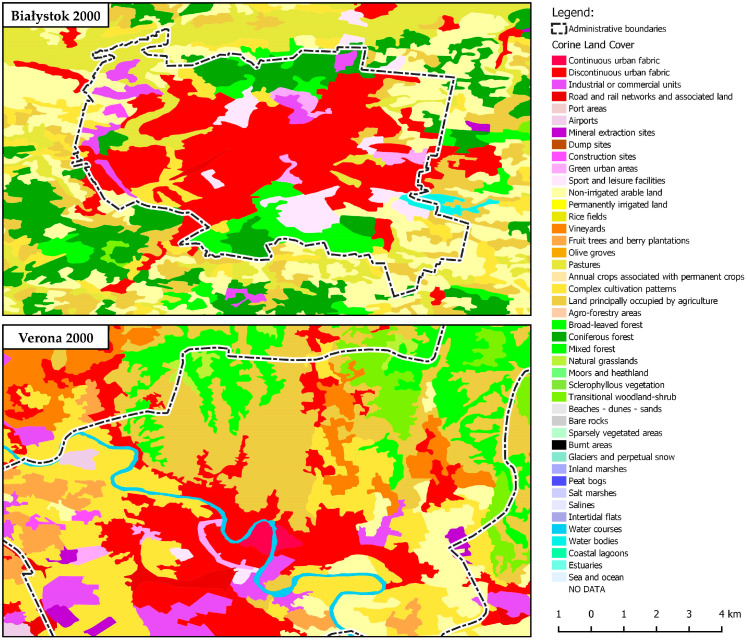
Land-use types identified on the basis of level 3 CLC data for selected cities.

**Figure 3 ijerph-20-02049-f003:**
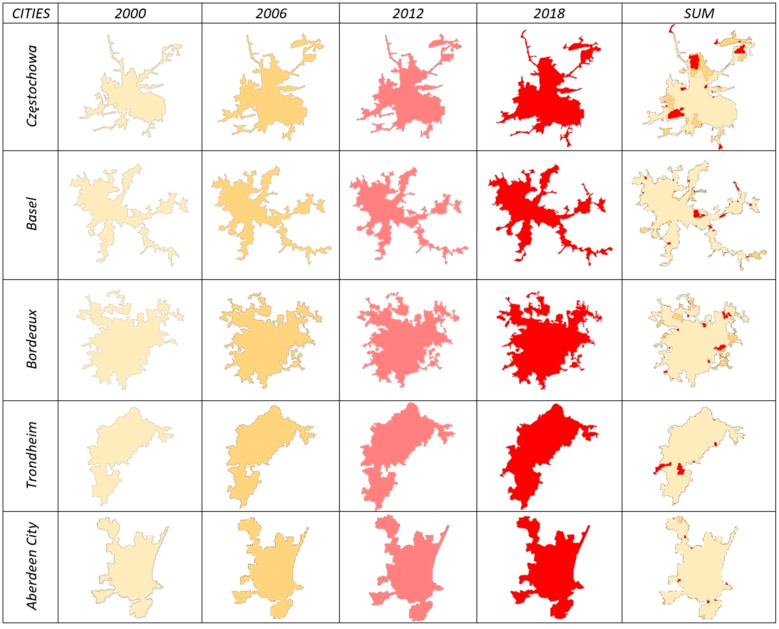
Changes in the urban boundaries of selected cities between 2000, 2006, 2012, and 2018.

**Figure 5 ijerph-20-02049-f005:**
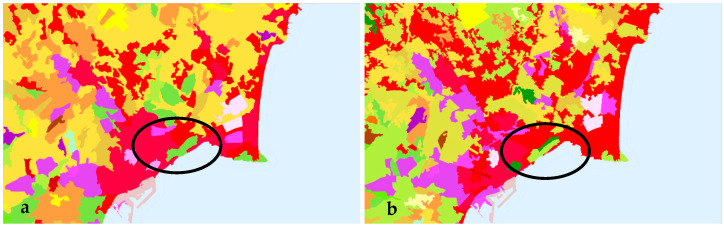
Classification of land-use types in Alicante: (**a**) 2000, (**b**) 2006.

**Figure 6 ijerph-20-02049-f006:**
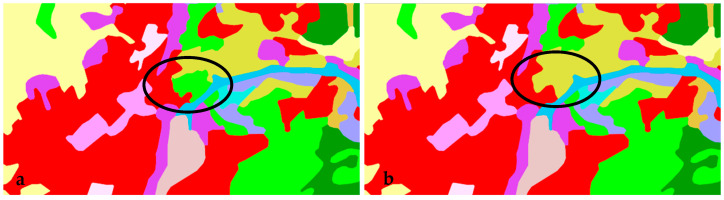
Classification of land-use types in Lubeck: (**a**) 2000, (**b**) 2006.

**Figure 7 ijerph-20-02049-f007:**
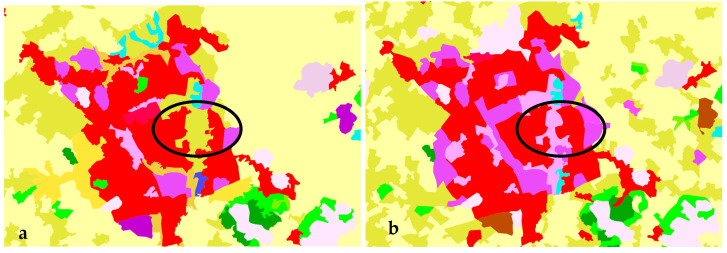
Classification of land-use types in Milton Keynes: (**a**) 2000, (**b**) 2006.

**Table 1 ijerph-20-02049-t001:** Urban expansion matrix for dividing the analyzed cities into eight classes. ↑ represents an increase in perimeter (*P*), area (*A*), or fractal dimension (*D_f_*) in the analyzed period; ↓ represents a decrease in perimeter, area, or fractal dimension in the analyzed period.

Class	*P*	*A*	*D_f_*
8	↑	↑	↑
7	↑	↑	↓
6	↑	↓	↑
5	↑	↓	↓
4	↓	↑	↑
3	↓	↑	↓
2	↓	↓	↑
1	↓	↓	↓

Note: The colours used in the table symbolize the different phases of spatial expansions, labelled as in Table 3 and Figure 4.

**Table 2 ijerph-20-02049-t002:** Increase in the area, perimeter, and fractal dimension of the analyzed cities.

No.	City	Δ*A* 2000–2006	Δ*P* 2000–2006	Δ*D_f_* 2000–2006	Δ*A* 2006–2012	Δ*P* 2006–2012	Δ*D_f_* 2006–2012	Δ*A* 2012–2018	Δ*P* 2012–2018	Δ*D_f_* 2012–2018
1	Aberdeen	0.034	0.059	0.002	0.017	−0.040	−0.004	0.022	0.004	0.003
2	Alicante	0.609	0.363	0.017	−0.051	−0.228	−0.023	0.034	0.051	0.006
3	Almeria	0.450	0.078	0.014	0.241	−0.131	−0.014	0.001	0.001	0.000
4	Augsburg	0.036	0.010	0.002	0.164	0.039	−0.013	0.003	−0.003	0.000
5	Basel	0.078	0.087	0.012	0.074	−0.046	−0.005	0.011	0.012	−0.007
6	Białystok	0.168	0.567	0.037	0.065	0.013	0.000	0.041	0.057	0.002
7	Bordeaux	0.043	0.032	0.004	0.112	0.003	0.000	0.010	−0.015	−0.001
8	Bournemouth	0.029	−0.046	−0.005	0.022	0.032	0.006	0.001	0.004	0.000
9	Brighton and Hove	0.044	−0.033	−0.003	0.021	−0.030	−0.002	0.010	0.028	0.003
10	Brunswick	0.056	0.090	0.009	0.222	−0.139	−0.019	0.002	0.000	0.000
11	Burgos	0.116	0.946	0.102	0.177	−0.042	−0.001	0.323	0.274	0.030
12	Cambridge	0.331	0.247	0.011	0.046	0.023	0.003	0.023	−0.017	−0.002
13	Cheltenham	0.037	0.112	0.013	0.016	−0.015	−0.003	0.008	0.025	0.003
14	Chemnitz	0.055	−0.042	−0.004	0.346	0.369	−0.004	−0.020	0.041	0.003
15	Colchester	0.149	−0.144	−0.021	0.026	−0.035	−0.003	0.019	0.011	0.002
16	Crawley	0.399	0.346	0.018	0.013	−0.001	0.000	0.003	0.002	0.000
17	Częstochowa	0.359	0.434	−0.014	0.014	0.016	0.002	0.002	0.000	0.000
18	Derby	0.009	0.118	0.019	0.002	0.083	0.008	0.019	0.049	0.006
19	Erfurt	0.101	0.168	0.014	0.072	0.045	0.007	0.010	0.020	−0.004
20	Freiburg im Breisgau	0.044	0.097	0.014	0.065	0.007	0.001	0.004	0.012	0.007
21	Geneva	0.158	0.050	0.009	0.023	0.024	0.003	0.013	0.006	−0.001
22	Gloucester	0.062	0.071	0.011	0.054	−0.078	−0.007	0.046	−0.010	−0.002
23	Osijek	0.004	−0.001	0.000	0.055	0.085	0.012	0.000	0.000	0.000
24	Rijeka	0.480	0.556	0.004	0.007	0.009	0.001	0.007	0.008	0.003
25	Split	0.324	0.494	−0.015	0.007	0.011	0.001	0.000	0.000	0.000
26	Graz	0.234	0.400	0.007	0.004	−0.007	−0.001	0.005	0.004	0.000
27	Karlsruhe	0.025	0.015	0.003	0.148	−0.063	−0.006	0.020	0.048	−0.001
28	Kiel	0.025	−0.002	0.001	0.093	−0.016	−0.010	−0.001	−0.018	0.002
29	Linz	0.082	0.070	−0.005	0.033	0.019	0.002	0.006	−0.003	0.000
30	Lubeck	0.033	0.075	0.007	0.115	−0.013	−0.004	0.029	0.016	0.001
31	Luton	0.010	−0.029	−0.004	0.004	0.025	0.004	0.006	0.007	0.000
32	Magdeburg	0.025	0.007	0.003	0.039	0.086	0.005	0.001	0.000	0.000
33	Messina	0.083	−0.060	−0.040	0.017	0.012	0.001	0.017	0.068	0.009
34	Milton Keynes	0.122	0.009	0.002	−0.001	0.071	0.008	0.023	0.038	0.005
35	Monchengladbach	0.007	0.001	0.000	0.579	0.362	−0.020	0.009	−0.003	−0.005
36	Munster	0.040	0.114	0.016	0.068	−0.007	0.020	0.000	0.000	0.000
37	Nantes	0.071	0.023	0.004	0.047	−0.038	−0.005	0.013	−0.009	−0.001
38	Newport	0.123	0.149	−0.016	0.030	−0.035	−0.004	0.002	0.008	0.001
39	Northampton	0.153	0.092	0.008	0.016	0.031	0.003	0.016	0.003	0.001
40	Norwich	0.179	0.014	0.005	0.035	0.067	0.005	0.019	0.028	0.004
41	Oviedo	0.336	0.226	0.019	0.154	−0.001	−0.006	0.006	0.016	0.001
42	Padua	0.081	0.087	0.011	0.065	−0.003	−0.001	0.019	0.005	−0.002
43	Radom	0.436	0.321	0.011	0.129	0.030	−0.001	0.023	0.024	0.002
44	Rennes	0.154	0.280	0.036	0.023	−0.009	−0.001	0.040	−0.033	−0.005
45	Rostock	0.059	0.059	0.010	0.120	−0.003	−0.004	0.003	−0.002	0.000
46	Salzburg	0.044	0.194	0.007	−0.019	0.040	0.004	0.000	0.006	0.001
47	Santander	0.259	−0.079	0.001	0.090	0.242	0.012	0.002	−0.003	0.000
48	Stavanger	0.020	0.029	−0.006	0.041	0.001	0.002	0.032	−0.016	0.001
49	Strasbourg	0.049	−0.003	0.003	0.021	0.029	0.001	0.002	−0.003	−0.001
50	Tampere	0.036	0.046	0.006	0.110	0.007	0.003	0.050	−0.037	−0.003
51	Taranto	0.014	0.038	0.003	0.028	−0.044	−0.002	0.026	−0.031	−0.004
52	Toruń	0.435	0.543	−0.002	0.093	0.094	0.002	0.003	0.002	0.000
53	Trieste	0.269	0.314	0.041	0.116	0.056	−0.008	0.009	0.015	0.001
54	Trondheim	0.016	0.067	0.010	0.078	0.022	0.000	0.010	0.002	0.001
55	Uppsala	0.018	−0.005	−0.001	0.036	0.054	0.010	0.005	−0.015	−0.001
56	Västerås	0.043	0.145	0.015	0.012	−0.006	−0.001	0.353	2.557	0.061
57	Verona	0.673	1.216	0.012	0.010	−0.001	0.000	0.029	0.106	−0.002
58	Vigo	0.226	1.338	0.056	0.018	−0.003	0.000	−0.240	−0.712	−0.052

**Table 3 ijerph-20-02049-t003:** Classification of cities in the analyzed time intervals.

No.	City	2006	2012	2018
1	Aberdeen	8	5	8
2	Alicante	8	1	8
3	Almeria	8	5	8
4	Augsburg	8	7	5
5	Basel	8	5	7
6	Białystok	8	8	8
7	Bordeaux	8	8	5
8	Bournemouth	5	8	8
9	Brighton and Hove	5	5	8
10	Brunswick	8	5	6
11	Burgos	8	5	8
12	Cambridge	8	8	5
13	Cheltenham	8	5	8
14	Chemnitz	5	7	4
15	Colchester	5	5	8
16	Crawley	8	5	8
17	Częstochowa	7	8	1
18	Derby	8	8	8
19	Erfurt	8	8	7
20	Freiburg im Breisgau	8	8	8
21	Geneva	8	8	7
22	Gloucester	8	5	5
23	Osijek	5	8	8
24	Rijeka	8	8	8
25	Split	7	8	1
26	Graz	8	5	8
27	Karlsruhe	8	5	7
28	Kiel	6	5	2
29	Linz	7	8	5
30	Lubeck	8	5	8
31	Luton	5	8	8
32	Magdeburg	8	8	5
33	Messina	5	8	8
34	Milton Keynes	8	4	8
35	Monchengladbach	8	7	5
36	Munster	8	6	1
37	Nantes	8	5	5
38	Newport	7	5	8
39	Northampton	8	8	8
40	Norwich	8	8	8
41	Oviedo	8	5	8
42	Padua	8	5	7
43	Radom	8	7	8
44	Rennes	8	5	5
45	Rostock	8	5	5
46	Salzburg	8	4	8
47	Santander	6	8	5
48	Stavanger	7	8	6
49	Strasbourg	6	8	5
50	Tampere	8	8	5
51	Taranto	8	5	5
52	Toruń	7	8	1
53	Trieste	8	7	8
54	Trondheim	8	7	8
55	Uppsala	5	8	5
56	Västerås	8	5	8
57	Verona	8	6	7
58	Vigo	8	5	1

Note: The colours used in the table symbolize the different phases of spatial expansions, labelled as in Table 1 and Figure 4.

## Data Availability

Data sharing not applicable.

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
