# Peer review of "An Analysis of the Spatial Development of European Cities Based on Their Geometry and the CORINE Land Cover (CLC) Database"

_ijerph, 2023, doi:10.3390/ijerph20032049_

Round 1

Reviewer 1 Report

The article "An analysis of the spatial development of European cities based on their geometry and the CORINE Land Cover (CLC) database" proposes to use the fractal dimension of the land area of 58 cities over 4 years to evaluate the type of growth of each of them.

After describing the indicators used and the data sets on which the study is based, the authors present the way in which the indicators are calculated, and then propose to classify the urban areas into 8 groups, corresponding to the gradients of the three chosen indicators (area, perimeter and fractal dimension).

The authors then present the results of their calculations for each city in the study in table and map form. They then discuss a few aspects, in order to illustrate the interest of the fractal dimension to better measure the extension of cities.

Some remarks on the form:
- The city of Nantes is missing in figure 1 (whose legend is on the next page)
- the use of colors to help visualize the classification is proposed from table 3. They could be used from table 1 to facilitate the reading
- the black and white printing makes it difficult to read this color gradient, the dark colors (green and red) being confused. This may be difficult to correct.
- I did not understand why more cities could not be selected, as it seems that the CLC data are larger

On the scientific content, I find the presentation of the measure, its calculation, and the discussions on its interest motivating. However, I would have liked to see more of what this tool allows. One could for example propose to compare the clustering (k-means type) of cities according to typical properties of its extension (area, number of inhabitants), then adding the perimeter and the fractal dimension, then discussing the differences in classification in order to illustrate the relevance of introducing this measure.
I find that the analysis lacks a step back to fully understand the interest of this measure.

Author Response

Response to Reviewer 1 Comments

First of all, we would like to thank you very much for the comments. We are convinced that your suggestions have significantly contributed to an improvement in our article. We do hope that we have succeeded in considering them in a satisfactory manner. Below are answers to specific comments.

- The city of Nantes is missing in figure 1 (whose legend is on the next page)

Response:

Suggestion has been inserted into the article. The figure 1 was corrected.

- the use of colors to help visualize the classification is proposed from table 3. They could be used from table 1 to facilitate the reading

Response:

Suggestion has been inserted into the article.

- the black and white printing makes it difficult to read this color gradient, the dark colors (green and red) being confused. This may be difficult to correct.

Response:

Suggestion has been inserted into the article.

- I did not understand why more cities could not be selected, as it seems that the CLC data are larger

Response:

The study was conducted on medium-sized European cities with a population of up to 300,000, which according to cited Urban agenda can be set an appropriate example of present trends in cities development. In addition, the use of data from the CLC database for the years 2000, 2006, 2012 and 2018, due to the different accuracy of the datasets, where older datasets have less accuracy than newer ones, required visual verification of the different land use types occurring for each urban area. The authors in this paper wanted to accurately determine geometric indicators for urban areas so they focused on 58 European cities, bearing in mind that historical CLC datasets have some limitations, such as the detailed nature of the input data and methods of interpretation.

Above explanations are included in the article in the line 239.

On the scientific content, I find the presentation of the measure, its calculation, and the discussions on its interest motivating. However, I would have liked to see more of what this tool allows. One could for example propose to compare the clustering (k-means type) of cities according to typical properties of its extension (area, number of inhabitants), then adding the perimeter and the fractal dimension, then discussing the differences in classification in order to illustrate the relevance of introducing this measure.
I find that the analysis lacks a step back to fully understand the interest of this measure.

Response:

Thank you for your valuable insights and comments. The research is part of a larger research project that will address the issues identified in the review. The authors believe that the research has justified the need to develop methodologies for research work on the subject of monitoring urban development.

                                                                                               Kind regards,

                                                                                                Szymon Czyża

Reviewer 2 Report

Based on CORINE land cover (CLC) data of 58 medium-sized European cities, this paper analyzes urban growth rate by fractal dimension, city area and city boundary length. It is innovative and helpful to determine the direction and speed of urban development in Europe. However, some problems still need to be corrected:

1. The background and hypothesis of the abstract are too long, and the final research conclusion need to promote. The eight development stages based on fractal dimension division are not shown.

2. The introduction should provide research progress and status, but the author spent a lot of time explaining the background, it should be revised.

3. In line 122-123, ‘These parameters are essential for determining a city's population which is one of the key variables in analyses of the spatial development of cities [38] and the dynamics of urban evolution’, the introduction of population factors is too blunt, it does not explain parameters, such as the length of urban boundaries, how reflect the population size.

4. It is suggested to improve the clarity of the picture, such as "Figure1" and "Figure2", Low clarity leads to a low recognition.

5. In the "study area" section, can you add some statements about the reasons for choosing these 58 cities, or the selection criteria?

6. ‘Figure 4. Classification of urban development stages’,There seems to be no difference with the simple description of words. Can you simplify the description of the program for calculating fractal dimension or change it directly to the description of words?

7. In line 392, It is necessary to add a separate explanation for the three indicators, such as what does "ΔDf increase" mean, and how will it lead to changes in urban morphology?

8. In line 481, a reasonable explanation should be given for the urban phenomenon like Alicant, which has a wide range of city grade changes. In line 500,the clear impact of CLC changes is not explained.

9. Figure 5. Phases of spatial expansion in the analyzed cities, The proportion difference in the figure is too large, the space expansion stage can not be well seen, it is suggested to enlarge.

10. All references need to be carefully checked, to correct the reference format.

Author Response

Response to Reviewer 2 Comments

First of all, we would like to thank you very much for the comments. We are convinced that your suggestions have significantly contributed to an improvement in our article. We do hope that we have succeeded in considering them in a satisfactory manner. Below are answers to specific comments.

1. The background and hypothesis of the abstract are too long, and the final research conclusion need to promote. The eight development stages based on fractal dimension division are not shown.

The proposed classification of development phases of medium-sized European cities indicates development trends and can be used as a tool for monitoring urban urbanisation. On the basis of the analysed cases, the stability of development of cities located in the central part of Europe can be observed. Also, we can find individual cases, such as Munster, Vigo, Kiel and Augsburg, where there is a steady decline in development phases. The authors point out that the usefulness of the tool will be even greater in the future, when it will be possible to carry out analyses for further time intervals. The methodology adopted also makes it possible to identify specific cases of changes in urban development phases using Alicante as an example (grades 8-1-8). Such cases require in-depth analyses. It should be noted that they may be due to the processes of intensive development of infrastructure investments taking place. Consequently, in their surroundings, many areas act as servicing areas for the investments, thus being classified as urbanised areas. On the other hand, after the investment is completed, they return to their original uses, being identified as non-urbanised areas. However, in the long term, the completed investment induces urbanisation processes in the neighbouring areas over time. Consequently, neighbouring areas are again identified as urbanised land use classes.

Above explanations are included in the article in the line 602.

2. The introduction should provide research progress and status, but the author spent a lot of time explaining the background, it should be revised.

This research is important because of the importance of monitoring urbanisation processes. The methodology adopted makes it possible to identify the trends and dynamics of the processes taking place, providing an important indicator for optimising the spatial policy of the cities under analysis. The proposed procedure may contribute to extending the scope of research in spatial terms (selection of analysed cities), temporal terms (successive time intervals), as well as in terms of indicators (the analysis may be extended by adding subsequent indicators, e.g. number of inhabitants). The application of new elements in the analysis may contribute to the enrichment of empirical research in the identification of urban development processes.

Above explanations are included in the article in the line 209.

3. In line 122-123, ‘These parameters are essential for determining a city's population which is one of the key variables in analyses of the spatial development of cities [38] and the dynamics of urban evolution’, the introduction of population factors is too blunt, it does not explain parameters, such as the length of urban boundaries, how reflect the population size.

Indeed, the indicated issue was not analysed. However, the authors assume that the issue raised in the review will be the subject of further research in the aspect of urban development.

4. It is suggested to improve the clarity of the picture, such as "Figure1" and "Figure2", Low clarity leads to a low recognition.

Response:

Suggestion has been inserted into the article. The Figure 1 and Figure 2 were corrected.

5. In the "study area" section, can you add some statements about the reasons for choosing these 58 cities, or the selection criteria?

The study was conducted on medium-sized European cities with a population of up to 300,000, which according to cited Urban agenda can be set an appropriate example of present trends in cities development. In addition, the use of data from the CLC database for the years 2000, 2006, 2012 and 2018, due to the different accuracy of the datasets, where older datasets have less accuracy than newer ones, required visual verification of the different land use types occurring for each urban area. The authors in this paper wanted to accurately determine geometric indicators for urban areas so they focused on 58 European cities, bearing in mind that historical CLC datasets have some limitations, such as the detailed nature of the input data and methods of interpretation.

Above explanations are included in the article in the line 239.

6. ‘Figure 4. Classification of urban development stages’, There seems to be no difference with the simple description of words. Can you simplify the description of the program for calculating fractal dimension or change it directly to the description of words?

Response:

Suggestion has been inserted into the article. According to suggestion authors change Figure 4 directly to the description of words.

Description is included in the article in the line 438.

7. In line 392, It is necessary to add a separate explanation for the three indicators, such as what does "ΔDf increase" mean, and how will it lead to changes in urban morphology?

Response:

The increase in the fractal dimension of cities' external boundaries is correlated with the rate of spatial development in urban areas. Higher values of increase in the fractal dimension imply a higher rate of spatial development, creating a ragged and dendritic urban fabric. On the other hand, the lower values of the increase indicator are mostly related to the filling the areas between the main traffic routes and areas vulnerable to the pressures of urbanisation processes while avoiding natural and anthropogenic obstacles.

Above explanations are included in the article in the line 356.

8. In line 481, a reasonable explanation should be given for the urban phenomenon like Alicant, which has a wide range of city grade changes. In line 500,the clear impact of CLC changes is not explained.

According to the research carried out, it can be concluded that there are specific cases like Alicante that require in-depth analysis. The background of the changes that have taken place is related to the development of transport infrastructure and the temporary identification of adjacent areas as built-up areas. The consequence of the processes that have occurred is the infilling of land located between the main traffic routes. Such processes affect the dynamics of change in terms of urban development phases. In view of the above, the authors believe that it makes sense to continue the research and extend it to include further time scales that would allow a more accurate identification of the urbanisation processes.

Above explanations are included in the article in the line 556.

9. ‘Figure 5. Phases of spatial expansion in the analyzed cities’, The proportion difference in the figure is too large, the space expansion stage can not be well seen, it is suggested to enlarge.

Response:

Suggestion has been inserted into the article. The Figure 5 was corrected.

10. All references need to be carefully checked, to correct the reference format.

Response:

All references were carefully checked and corrected.

                                                                                               Kind regards,
                                                                                                Szymon Czyża

Round 2

Reviewer 2 Report

(1)please check: [12,13][14][15][16][17] in line 52ï¼›[46][6,11,12,37,40,47–50][51] in line 115ï¼›[60][61,66–71] in line 143ï¼›they have clerical errorï¼›

(2)CLC data in 190 should indicate which year.

Author Response

Response to Reviewer 2 Comments

Once again, we would like to thank you very much for the comments. We are convinced that your suggestions have significantly contributed to an improvement in our article. We hope that we have succeeded in considering them in a satisfactory manner. Below are answers to comments.

(1)please check: [12,13][14][15][16][17] in line 52ï¼›[46][6,11,12,37,40,47–50][51] in line 115ï¼›[60][61,66–71] in line 143ï¼›they have clerical errorï¼›

Errors has been corrected and all references were carefully checked.

(2)CLC data in 190 should indicate which year.

Suggestion has been inserted into the article.

                                                                                                Kind regards,

                                                                                                Szymon Czyża
